# The effects of mating and blood feeding on the immune defense of female *Aedes aegypti* mosquitoes

**Brendan J. Kelly**[1,2°]**, Nicki C. Joseph**[1°]**, Dom Magistrado**[1]**, Sarah M. Short**[1*]

**1** Department of Entomology, The Ohio State University, Columbus, Ohio, United States of America,
**2** University of Massachusetts-Amherst, Amherst, Massachusetts, United States of America

☯ These authors contributed equally
* short.343@osu.edu

## Abstract

Mating and consequent reproduction have variable effects on the immune defenses of female insects, ranging from immune enhancement to immune suppression. Blood feeding, which is necessary to induce egg production in anautogenous insect species, also impacts immunity, with variable effects in different insect species. Because immunity is a key component of vector competence, and mating and blood feeding can influence immunity, we investigated the effects of these processes on immune defense in the yellow fever mosquito, *Aedes aegypti*. To explore both aspects of reproduction, we measured survival, bacterial load, and bacterial prevalence in response to mating and blood feeding in adult female *Ae. aegypti* infected with various doses of the opportunistic bacterial pathogen, *Serratia marcescens*. When we challenged females with a high dose of *S. marcescens*, we found that mating induced significantly higher survival probabilities and bacterial loads in non-blood fed individuals. Mating improved survival after infection with a moderate dose of *S. marcescens* as well, but when we challenged females with a low dose, we observed no effect of mating on survival or bacterial load. Thus, the effects of mating on immune defense appear to be dose dependent. Blood feeding, on the other hand, consistently reduced survival regardless of dose. By exploring how key life history traits impact immune defense, our results continue to advance a thorough understanding of how *Ae. aegypti* defend against infection.

## Author summary

Mating and the subsequent process of reproduction can either enhance or suppress the ability of female insects to respond to pathogenic infection. In most mosquitoes, blood feeding is also required for egg production and has been shown in multiple species to impact immunity. Because immunity can affect whether mosquitoes transmit pathogens to mammals and because mating and

**Data availability statement:** Raw data for all experiments are contained within S1 Dataset.

**Funding:** BJK, NCJ, DM, and SMS were supported by the Ohio State University Infectious Diseases Institute (https://idi.osu.edu/) and the Ohio State University College of Food, Agricultural, and Environmental Sciences (https://cfaes.osu.edu/). This material is also based upon work supported by the National Science Foundation Graduate Research Fellowship Program under Grant No. DGE-1343012 awarded to DM. Any opinions, findings, and conclusions or recommendations expressed in this material are those of the author(s) and do not necessarily reflect the views of the National Science Foundation. The funders had no role in study design, data collection and analysis, decision to publish, or preparation of the manuscript.

**Competing interests:** The authors have declared that no competing interests exist.

blood feeding are a key part of female mosquito life history, we investigated how mating and blood feeding impact immune defense in the yellow fever mosquito, *Aedes aegypti*. We found that mating generally improved the ability of female *Ae. aegypti* to survive pathogenic bacterial infection while blood feeding generally decreased their ability to survive. We also found that the effect of mating was dose dependent while the effect of blood feeding was not. Our findings suggest that the understanding of immune defense in *Ae. aegypti* can be improved by examining physiological processes that have the potential to interact with and shape the mosquito's response to infection.

## 1. Introduction

The yellow fever mosquito, *Aedes aegypti*, transmits multiple human pathogens, including dengue, chikungunya, and Zika viruses [1,2,3]. Like most insects, female *Ae. aegypti* must mate to reproduce. Mating has profound effects on female *Ae. aegypti* physiology, with some having the potential to impact successful transmission of pathogens (vectoral capacity). For example, mating significantly increases *Ae. aegypti* lifespan, which is a primary determinant of vectorial capacity [4,5]. Mating also decreases biosynthesis of Juvenile Hormone (JH) III in blood fed (BF) [but not in non-blood fed (NBF)] individuals, and JH has been shown to be immunosuppressive [6,7,8]. Additionally, mating affects immune gene activity; multiple transcripts linked to immunity have a ~10-fold mean increase in expression 24 hours post-mating in the reproductive tract [9]. Mating has also been shown to affect bacterial load in *Ae. aegypti.* Mated females infected intrathoracically with *Escherichia coli* present with significant increases in bacterial load 24 hours post-infection compared to virgin females [10]. Since immune defense is a key determinant of vector competence and therefore pathogen transmission in *Ae. aegypti* [11], it is imperative to understand how mating impacts immune defense in this system.

Mating has a variety of effects on the immune defense and immune system activity of female insects, from immune suppression to immune enhancement [12–18]). Multiple studies report that mating leads to a decrease in female immune capacity, a phenomenon referred to as "post-mating immune suppression." For example, female *Drosophila melanogaster* infected with bacterial pathogens display decreased survival and increased bacterial loads following mating [19,14]. It has also been shown that when sex peptide is transferred from male to female *D. melanogaster* during mating, JH synthesis is induced, which subsequently suppresses immune defenses in females post-mating [20]. In mealworm beetles, *Tenebrio molitor*, mating leads to decreased phenoloxidase activity (a critical component of the insect immune system) in both males and females [12]. Mated females of the mosquito *Anopheles coluzzii* show increased susceptibility to the malaria parasite *Plasmodium falciparum* compared to virgin females [15]. These results illustrate that post-mating immune suppression is observed among a diversity of insect systems, including mosquitoes, and in response to a diversity of pathogens.

Mating and immune defense are not always antagonistic to each other, however. There have been several examples of post-mating immune enhancement, i.e., an increase in immune capacity, following mating. In Texas field crickets, *Gryllus texensis*, mated females are able to survive infection with the bacteria *Serratia marcescens* better compared to virgin females [16]. Mated buff-tailed bumblebee females, *Bombus terrestris*, are less likely to become infected with the trypanosomal parasite *Crithidia bombi* compared to females given access to, but prevented from mating with, males [17]. Even though mating decreases phenoloxidase activity in *T. molitor* [12], mating leads to improved survival in response to infection with *Beauveria bassiana*, a fungal pathogen [18]. Together, these results provide evidence that post-mating immune enhancement is also observed in different insect taxa and in response to a variety of pathogens and parasites.

In *D. melanogaster* and many other insect species, mating alone is sufficient to induce egg production. However, in *Ae. aegypti* and other anautogenous insect species, reproduction requires both mating and blood feeding, as females must blood feed to develop mature oocytes [21]. Because mating and blood feeding are inextricably related to each other within the context of mosquito reproduction, it is relevant to measure the effects of blood feeding in tandem with the effects of mating on mosquito immune defense. Blood feeding has dramatic effects on aspects of mosquito physiology, including some aspects that are related to immune defense [22–25]. Among *Anopheles* species, blood feeding generally improves immune capacity. For example, blood feeding increases melanization efficacy, results in a proliferation of hemocytes, and induces increased expression of proteins produced by hemocytes [22,24]. However, effects of blood feeding in *Ae. aegypti* are more varied. Blood feeding increases hemocyte number in *Ae. aegypti* [26] but also leads to a downregulation of many immune genes [23]. Further, a microRNA that is only induced upon blood feeding in *Ae. aegypti* suppresses the Toll pathway, a component of the immune response, in vivo and in vitro [27]. In *Ae. aegypti* infected with *E. coli,* blood feeding generally decreases bacterial load early in infection and corresponds with a decrease in survival [26,28]. However, these effects are dose dependent, with reductions in survival only seen at high doses. Because immunity is a key component of vector competence, and mating and blood feeding can influence immunity, we are interested in investigating the effects of these processes on immune defense in *Ae. aegypti*. Specifically, we are focused on the effects that environmental pathogens such as *S. marcescens* may have on *Ae. aegypti.* While not all female *Ae. aegypti* will become vectors of human pathogens in their lifetime, all will encounter environmental bacteria, which influence mosquito life history and have the potential to affect human pathogen transmission. To explore the effects of mating and blood feeding on immune defense, we measured survival and bacterial load in response to mating and blood feeding treatments in 5–6 day old female *Ae. aegypti* infected with different doses of the opportunistic bacterial pathogen *S. marcescens*. Survival and bacterial load are both whole-organism measures of overall infection outcome and are henceforth collectively referred to as measures of "immune defense." We chose to use *S. marcescens* because it commonly infects insects and can cause insect mortality, including in *Ae. aegypti* [29,30,31].

## 2. Methods

### 2.1. Experimental design

To assess survival and bacterial load in response to mating and blood feeding treatments in *Ae. aegypti* females, we conducted four experiments. In all experiments, females were 3–4 days old when mated, 4–5 days old when blood fed (where applicable) and 5–6 days old at the time of infection. High dose experiment: We infected BF and/or NBF Liverpool IB12 (LVP) [32] strain females with a high dose of *S. marcescens*, and examined survival, bacterial load (the total quantity of bacteria in an individual), and infection prevalence (the percentage of infected individuals in a treatment group) over the course of seven days. High vs. moderate dose experiment: We infected NBF LVP strain females with high and moderate doses of *S. marcescens* and examined survival over the course of 24 hours. Low dose experiment (LVP): we infected NBF and BF LVP females with a low dose of *S. marcescens*, examined survival over the course of 24 hours, and assessed bacterial load and infection prevalence at 16 hours post-infection (Fig 1). Moderate dose experiment

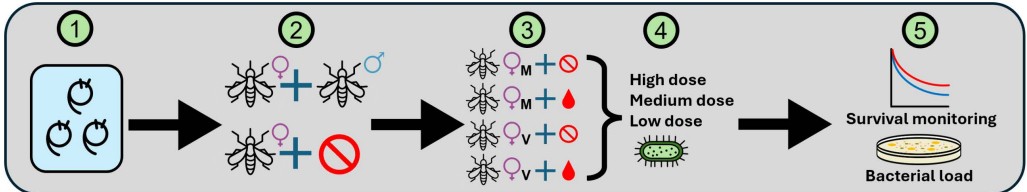

**Fig 1. Experimental overview.** Experimental setup for LVP strain females with steps labeled in green. Pupae were reared to adulthood, and males/females were separated shortly after emergence to ensure virgin status (step 1). Half of the females were then mated to males while the rest remained virgin (3-4 days old; step 2). A subset of mated and virgin individuals were then blood fed while the rest were maintained on 10% sucrose (4-5 days old; step 3). Females were then infected with varying doses of *S. marcescens* (low, moderate, or high) via pinprick infections (5-6 days old; step 4), and were then monitored for survival and bacterial load post-infection (step 5). This figure was created manually using Microsoft PowerPoint for Microsoft 365 MSO (Version 2505 Build 16.0.18827.20102) 64-bit.

(THAI): A fourth moderate dose infection experiment using NBF THAI strain females [5] was conducted to examine the effect of mating on survival and bacterial load in individuals more closely resembling field populations. The methods and data for this experiment can be found in S1 Protocol and S1 Fig, respectively.

## 2.2. Mosquito rearing

We reared *Ae. aegypti* mosquitoes at 27°C, 80% relative humidity, and a photoperiod of 14 hours light: 10 hours dark. Eggs were hatched under a vacuum, and populations were thinned to 200–300 first instar larvae per tray. Larvae were provided cat food (9Lives, Indoor Complete) *ad libitum* until pupation at which point they were placed into a cage with *ad libitum* access to 10% sucrose (w/v) for emergence.

## 2.3. Adult collection and separation by sex

We collected adults within 12–14 hours of emergence to ensure virgin status [33,34,35] using a mechanical aspirator (BioQuip Products, Inc.). We sorted mosquitoes by sex and kept males in 1 pint paper cups in groups of 25 individuals. Females were kept in 2.5L plastic cages in groups of approximately 50 individuals. All adults were given *ad libitum* access to 10% sucrose and aged 3–4 days before proceeding with treatment.

We conducted a preliminary experiment to verify that collecting females 12–14 hours post-emergence would ensure virgin status. We dissected spermathecae from ~85 LVP and THAI strain females over three replicate experiments and checked for the presence of sperm under a compound microscope. No sperm were present in any individuals from either strain, confirming that virgin status was maintained in females collected 12–14 hours post-emergence (S2 Fig).

## 2.4. Mating procedure

Starting 0–1 hour before incubator dawn, we introduced approximately 75 males to cages of 50 females to allow for mating. Females aged 3–4 days were allowed to mate with males for 30–36 hours. Directly after mating, we cold anesthetized each group to sort females from males. We also cold anesthetized females from the virgin treatments to control for the effect of cold shock. We then returned each group of females to their 2.5L cage with *ad libitum* access to 10% sucrose.

We conducted a preliminary experiment to verify that 30–36 hours was sufficient for all females to mate. We dissected spermathecae from ~50 LVP and THAI strain females over two replicate experiments and checked for the presence of sperm under a compound microscope. Sperm were present in all individuals from each strain, thus confirming that 30–36 hours was sufficient for mating to take place (S3 Fig).

## 2.5. Blood feeding procedure

Prior to completion of the mating procedure (i.e., 26–32 hours after the start of mating), a subset of virgin and mated females were blood fed. To blood feed mosquitoes, we prepared blood meals consisting of rabbit's blood (HemoStat Laboratories) and 5′-ATP-Na$_2$ (Sigma-Aldrich) at a concentration of 10 µM 5′-ATP-Na$_2$/mL rabbit's blood. We added approximately 3 mL of this bloodmeal to a Hemotek reservoir (Hemotek) wrapped in parafilm (Bemis) and, using the Hemotek, warmed the blood meals to 37°C. We allowed 4–5 day-old females to feed for 1–2 hours. At the end of the mating procedure, when all females were cold anesthetized to remove males, we also removed females that were provided a blood meal but did not feed. We considered "blood fed" to include any females that had visibly taken a bloodmeal, including partially engorged females.

## 2.6. Infection with *S. marcescens*

Approximately 24 hours after blood feeding, we infected females from all treatment groups with *S. marcescens.* For all infection experiments, we used a strain of *S. marcescens* transformed with pPROBE-Kan with *PnptII-gfp* fusion [36,37]. Our strain of *S. marcescens* was therefore kanamycin resistant and expressed Green Fluorescent Protein (GFP), which allowed us to verify which bacteria were *S. marcescens* when assaying bacterial load.

We infected 5–6 day old mosquitos using intrathoracic infections based on the method described by Khalil et al. [38]. Briefly, we fixed a 0.2mm Minutien pin (Austerlitz Insect Pins) to the end of a 200 µL plastic pipette tip (VWR). To prepare the bacterial culture used during infection, we cultured *S. marcescens-GFP* in lysogeny broth (LB) plus kanamycin (50 µg/mL) at 30°C with 220 RPM shaking for 16–18 hours. After incubation, we diluted the culture to 1.0 O.D.$_{600}$, and then diluted this 1.0 O.D.$_{600}$ culture with sterile 1X PBS by 1:5. The only exception to this procedure was to generate a "high" dose for the High vs. Moderate dose experiment. In this experiment, the high dose was generated by using an undiluted 1.0 O.D.$_{600}$ culture. To infect, we dipped the pin into the bacterial culture, and punctured the cold anesthetized female's thorax, dipping the pin again in culture between females. We performed sterile injury controls to ensure that puncturing alone was not causing mortality to the females. To do so, we dipped the pin in 70% ethanol to sterilize, and then into sterile 1X PBS before puncturing the thorax. We observed no more than 1% mortality in any treatment group after sterile injury. These data can be found in S1 Dataset.

For reasons that are unclear, the average dose delivered by intrathoracic infections was variable between experiments, ranging from 9.38 ± 1.17 colony forming units (CFU)/mosquito to 113.16 ± 15.92, resulting in delivery of high, moderate, and low numbers of bacteria, which we henceforth refer to as "high," "moderate" and "low" doses. To account for and document this variability, we quantified the average infectious dose for each experiment after removal of outliers (identified as points more than 1.5 times above or below the interquartile range), and have indicated this throughout the results. Dosing data for all experiments can be found in S2 Dataset.

## 2.7. Survival and bacterial load monitoring

For bacterial load measurements, we homogenized individual whole mosquitoes 16 hours post-infection in tubes of 150 µL of sterile 1X PBS with a mechanical homogenizer (VWR). We serially diluted homogenates and spread 100 µL of each dilution and the undiluted homogenate on LB + kanamycin (50 µg/mL) agar media. Plates were then left to incubate at room temperature for approximately 48 hours, or until bacterial load was countable under a NightSea light and filter adapter set-violet. Following the incubation period, green fluorescent colonies on each plate were manually counted. In cases where colonies were very numerous, we estimated the density of the plate by counting ¼ of the plate and multiplying by four. For the high dose experiment, samples where the density of bacteria prohibited counting were given an estimated value of the highest countable plate.

For the high dose experiment, we collected samples for bacterial load measurement at days zero, one, and seven. For the low dose experiment, we collected samples for bacterial load at hour zero and 16.

After injections, survival was manually monitored by counting the number of females that failed to respond to physical stimuli (e.g., tapping the cage). Dead females were removed from the cage to ensure following deaths were not a result of environmental bacteria exposure.

### 2.8. Statistical analysis

All analyses were performed using RStudio version 4.4.0 and R version 4.3.3 and *PSCL*, *lmtest*, *survminer* and *survival* packages [39–47].

All survival data were analyzed using Cox proportional hazards models including blood feeding status, mating status, and/or dose as predictor variables where appropriate. To analyze bacterial load data, we used different approaches depending on the nature of each dataset. In datasets with an excess of zero values, we used a hurdle model with a negative binomial distribution to simultaneously analyze the effects of mating status and day on the presence/absence of bacteria as well as CFU count data. For all other bacterial load analyses, non-zero count data were analyzed using a linear model to assess the effects of blood feeding status and/or mating status as appropriate on bacterial load, and presence/absence data were analyzed using a generalized linear model (GLM) with a binomial distribution. The statistical approach used for each analysis is indicated in each respective figure legend. For all analyses, a backwards elimination approach was taken to obtain the final model and assess significance. Presence/absence box plots were all created using Microsoft Excel for Microsoft 365 MSO (Version 2405 Build 16.0.17628.20006) 64-bit. Our experimental design figure was created manually using Microsoft PowerPoint for Microsoft 365 MSO (Version 2505 Build 16.0.18827.20102) 64-bit. All other graphs were created using R version 4.3.3. and RStudio version 4.4.0. All raw data, R code, and model outputs can be found in S1 Dataset, S1 Code, and S2 Code respectively.

## 3. Results

### 3.1. High dose infections

We tested whether mating, blood feeding status, or an interaction between these two factors significantly affected the survival of LVP strain females after infection with a high dose of *S. marcescens* over seven days ($113.16 \pm 15.92$ CFU injected per female) (Fig 2A). We found that the interaction between blood feeding and mating was non-significant ($p = 0.065$), while blood feeding significantly lowered survival ($p = 2.324 \times 10^{-07}$) and mating significantly improved survival ($p = 0.003$) post-infection. When we parsed the dataset by blood feeding status, we found that in NBF females, mating significantly increased survival ($p = 0.002$), while in BF females it had no effect ($p = 0.325$). We also examined bacterial load and infection prevalence at days 1 and 7 post-infection (Figs 2B and 2C). This was necessarily limited to NBF females because high mortality among BF females prevented us from collecting bacterial load data. We tested the effects of mating, day, replicate, and an interaction between mating and day on bacterial count data (Fig 2B) and presence/absence data (Fig 2C) among NBF females. Bacterial count analysis revealed a significant effect of the interaction between mating and day on bacterial load ($p = 1.838 \times 10^{-06}$). We then parsed the dataset by day and performed subsequent analyses to assess the effect of mating on each day. We found that mated females had significantly higher bacterial loads than virgin females on day one ($p = 1.168 \times 10^{-4}$), but mating had no effect on day seven ($p = 0.225$). Presence/absence data analysis revealed no significant effects of mating, day, or the interaction between these factors on the prevalence of infection (Fig 2C).

### 3.2. High vs. moderate dose infections

We then infected mated and virgin NBF LVP females with high ($108.64 \pm 13.31$ CFU) and moderate ($42.33 \pm 9.76$ CFU) doses of *S. marcescens* and monitored the survival of each group (Fig 3). We found no significant interaction between dose and mating status ($p = 0.864$). Mating significantly increased survival ($p = 0.009$), and dose also significantly affected

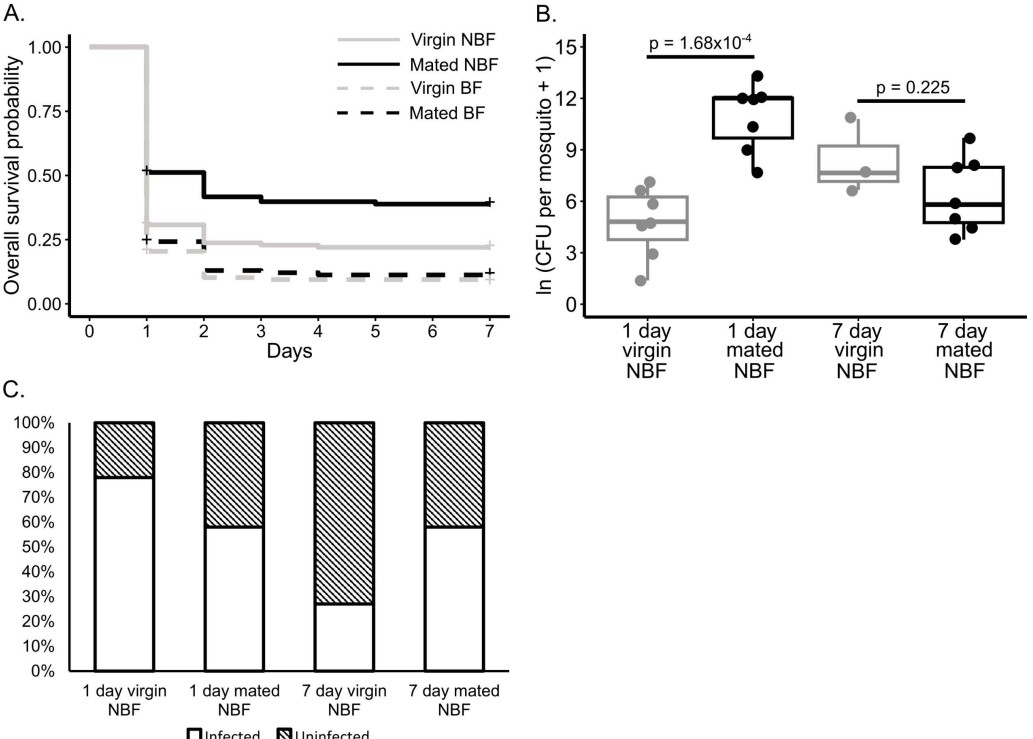

**Fig 2. Mating significantly increases survival and bacterial load in females infected with a high dose of *S. marcescens*.** Survival and bacterial load of LVP females injected with 113.16 ± 15.92 CFU of *S. marcescens*. The full experiment was replicated four times. **A.** Survival probability of virgin (gray line) and mated (black line) LVP females either BF (dashed lines), or NBF (solid lines) over seven days (virgin NBF n = 137, mated NBF n = 143, virgin BF n = 149, mated BF n = 64). A Cox proportional hazards model showed no significant interaction between mating and blood feeding status (p = 0.065). Both blood feeding (p = 2.324 x 10^-07) and mating status (p = 0.003) significantly predicted survival. Mating significantly increased survival in NBF females (p = 0.002), while in BF females, it had no effect (p = 0.325). **B.** Count data (i.e., non-zero CFU values) for virgin (gray) and mated (black) NBF LVP females at one and seven days post-infection. Using a hurdle model, we found a significant interaction between mating and day (p = 1.838 x 10^-06) for count data. A significant effect of mating was found for day one (p = 1.168 x 10^-4), but not for day seven (p = 0.225). **C.** Infection prevalence for NBF virgin and mated females one and seven days post-infection (n = 9-13 per experimental group). Uninfected is represented by black stripes and infected by no stripes. A hurdle model analysis on presence/absence data revealed no significant effects of mating, day, or the interaction between these factors on infection prevalence.

survival (p = 2.482 x 10^-12), with the moderate dose corresponding to higher survival post-infection (Fig 3). When we infected THAI strain females in a separate experiment with a similar moderate dose infection (S1 Fig), we found that, unlike LVP, mating did not significantly affect survival rate after infection (p = 0.238, S1A Fig). We also found that mating status was not a significant predictor of bacterial load (p = 0.316, S1B Fig) nor prevalence of infection (p = 0.331, S1C Fig) in THAI females at this dose.

### 3.3. Low dose infections

Finally, we tested the effects of mating and blood feeding on female survival and bacterial load using a low dose of *S. marcescens* (9.38 ± 1.17 CFU per female). We found no significant interaction between mating and blood feeding (p = 0.251) and no main effect of mating on survival (p = 0.756, Fig 4A). However, blood feeding did significantly reduce survival (p = 0.003, Fig 4A). Analysis of bacterial load count data in LVP revealed that there was no significant interaction between mating and blood feeding on bacterial load (p = 0.624, Fig 4B). Moreover, neither mating status (p = 0.359) nor blood feeding status (p = 0.525) had a significant effect on bacterial load count data (Fig 4B). Analysis of presence/

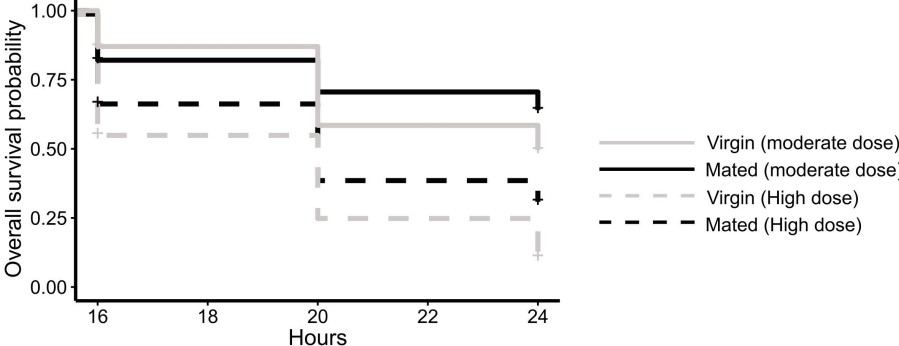

**Fig 3. Mating significantly increases survival in LVP females infected with a moderate dose of *S. marcescens*.** Survival probability of NBF LVP females injected with a high or moderate dose of *S. marcescens.* The moderate dose (42.33±9.76 CFU/female) was significantly lower than the high dose (108.64±13.31 CFU/female) as measured via t-test (t=4.02, p=2.03 x 10⁻⁴). The full experiment was replicated four times and overall sample sizes are n=81-87 per group. Gray lines represent virgin females, and black lines represent mated females. Solid lines depict the moderate dose, and dotted lines represent the high dose. A Cox proportional hazards model showed no significant interaction between dose and mating (p=0.864) but did reveal a significant main effect of both dose (p=2.482 x 10⁻¹²) and mating (p=0.009) on survival.

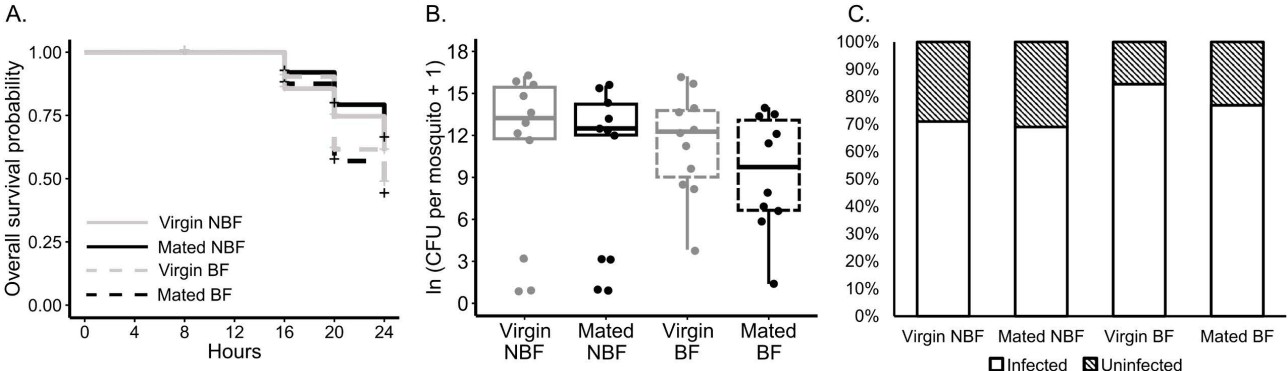

**Fig 4. Mating status does not affect female survival or bacterial load in females infected with a low dose of *S. marcescens*.** Survival and bacterial load of females injected with 9.38±1.17 of *S. marcescens*. **A.** Survival of virgin (gray line) and mated (black line) females either BF (dashed lines), or NBF (solid lines) following infection with *S. marcescens* (virgin NBF n=137, mated NBF n=143, virgin BF n=149, mated BF n=64; data collected over 4 replicate experiments). A Cox proportional hazards model showed no significant interaction between mating and blood feeding status (p=0.251). No significant effect of mating was found (p=0.756), but there was a significant effect of blood feeding on survival (p=0.003). **B.** Count data (i.e., non-zero CFU values) of virgin (gray) and mated (black) females at 16 hours post-infection. A linear model showed no significant interaction between mating and blood feeding status (p=0.624), and no significant main effect of blood feeding (p=0.525) or mating status (p=0.359) on bacterial load. **C.** Infection prevalence of NBF females (n=13-14 for all groups); uninfected is represented by black stripes and infected by no stripes. A GLM showed no significant interaction between mating and blood feeding status (p=0.751) and no significant effect of mating status (p=0.623), or blood feeding status (p=0.360) on infection prevalence at 16 hours post-infection.

absence data showed no significant interaction between mating and blood feeding on infection prevalence (p=0.751), and neither mating status (p=0.623) nor blood feeding status (p=0.360) significantly predicted infection prevalence (Fig 4C).

## 4. Discussion

In the current work, we demonstrated that mated LVP *Ae. aegypti* females showed significantly higher survival probabilities and higher bacterial loads compared to virgin females when challenged with a relatively high dose of the bacterial pathogen *S. marcescens*. Similarly, at a relatively moderate infection dose, survival was significantly improved by mating

in LVP strain females. However, the effect of mating on survival and bacterial load was lost in LVP females after infection with a relatively low dose. Thus, the effect of mating on survival and bacterial load in LVP strain females appears to be dose dependent.

Our results are consistent with those of Reitmayer et al. [10] in which mating was reported to significantly increase *E. coli* bacterial load in non-blood fed *Ae. aegypti* females 24 hours post-infection. However, we also found that mating significantly improved survival after infection. This juxtaposition between survival and bacterial load in non-blood fed, mated individuals suggests that mating may have an effect on infection tolerance (i.e., ability of an organism to remain healthy after an infection) [48], but this possibility requires further investigation.

Mating has been reported to have diverse impacts on infection outcomes in other systems. For example, mating increases survival after pathogenic infection in both *Gryllus* crickets [16] and *Tenebrio* beetles [18]. However, in many dipteran species (e.g., *Drosophila melanogaster* and *Anopheles coluzzii*), mating causes immunosuppression [14,15]. The fact that relationships between immune defense and mating are species dependent is potentially linked to reproductive hormones which are differentially transferred, synthesized, and/or utilized during and after mating. In *D. melanogaster*, synthesis rates of JH increase in females post-mating due to the transfer of seminal fluid proteins [49]. This mating-induced increase in JH production subsequently increases susceptibility to infection, likely through decreased antimicrobial peptide activity [20]. Unlike *Drosophila*, however, the relationship between JH, mating, and the immune response in *Ae. aegypti* is less clear. Similar to *D. melanogaster,* JH reduces expression of antimicrobial peptide genes in *Ae. aegypti* [7]. However, mating has no effect on JHIII biosynthesis in NBF *Ae. aegypti,* [6]. This, combined with the findings of the present study, suggest that the effects of mating on immune defense in NBF female *Ae. aegypti* do not follow the same mechanistic paradigm as that in *D. melanogaster*. However, many uncertainties remain.

Male *Ae. aegypti* transfer JH to females during mating [50], but it is unclear how this process may potentially influence the local immune response or systemic immune defense. JH titers also have the potential to differ across tissues that are impacted by JH, i.e., the ovaries, fat body and midgut [51,8]. Localized increases in JH could cause tissue-specific immune responses that we would not have necessarily detected here.

Another key consideration regarding JH, reproduction, and immunity is that anautogenous mosquitoes like *Ae. aegypti* require a blood meal to produce eggs for each gonotrophic cycle, so that the relationship between reproduction and immunity necessarily includes any effect of blood feeding. In *Ae. aegypti*, JH synthesis displays a cyclical pattern post-blood feeding; JH titers decrease immediately after blood feeding, remain low for approximately 48 hours and then increase after the onset of vitellogenesis [52]. We found that blood feeding dramatically reduced immune defense, and that mating had no effect on immune defense in BF females. Moreover, we observed this at a time post-blood feeding when JH levels are expected to be relatively low, egg production is expected to be relatively high compared to NBF females (12–48 hours post-blood feeding, [52]), and at a time when JH levels are expected to be lower in mated BF females relative to virgin BF females [6]. This, coupled with the fact that JH has been shown to be immunosuppressive [7,8], suggests it is unlikely that JH is driving the effects of blood feeding on immune defense and lends further credence to our earlier assertion that JH is not mediating the effects of mating on immune defense. It is possible that other aspects of reproduction such as ejaculatory components transferred through mating (e.g., seminal fluid proteins;[53]) or physiological processes related to egg production may be causing these effects. We also note that, given the dynamic nature of hormone production as it relates to mating and blood feeding in mosquitoes [52], an infection initiated at a different time point relative to mating or blood feeding may result in a very different outcome.

Our results indicate that mating improves survival after infection, and are another example of a benefit that female *Ae. aegypti* experience from mating. Previous work has shown that mating improves the longevity of *Ae. aegypti* females absent of infection [5]. As *Ae. aegypti* are primarily monandrous in their mating behavior [54], it may be evolutionarily advantageous for males to promote the health of their mates [13]. Our results are consistent with this hypothesis at high and moderate dose infections, as mating does in fact correspond to a significant increase in survival probability for

infected females. However, it remains to be seen how the improvements to infection survival we report here potentially impose costs and/or tradeoffs to reproduction and therefore impact lifetime reproductive fitness. Additionally, because the effect of mating is lost with a decrease in dose, it may be inferred that the effect of mating on immune enhancement is only evidenced in the case of an immune challenge of moderate to high virulence.

Our results suggest that mating impacts survival and bacterial load only in *Ae. aegypti* which have not blood-fed. In blood-fed females, we observed no effect of mating. We also observed that blood feeding reduced survival post-infection, regardless of mating status or dose. The effects of blood feeding on immune defense can vary in different mosquito species. In both *An. gambiae* and *Ae. aegypti*, blood feeding leads to the proliferation of hemocytes, which mediate the cellular immune response [26,24]. *An. stephensi* show a similar pattern, as blood feeding leads to an improved melanization response in females provided with sufficient sucrose [22]. In our study, however, blood feeding consistently led to a decrease in survival in females injected with high and low doses of *S. marcescens*. These results are consistent with previous studies in *Ae. aegypti*. Blood feeding can decrease bacterial load early in infection in *Ae. aegypti* infected with *E. coli* [26,28] but corresponds with a decrease in survival [26] . Notably, blood feeding does not affect survival after infection with low doses of *E. coli* (i.e., $1 \times 10^3$ CFU or lower) [26,28]. We observed an effect of blood feeding on survival at a low dose (i.e., 9.38 ± 1.17 CFU), suggesting the effect of blood feeding on infection survival may vary across pathogens or may depend not on dose per se but rather on overall virulence of infection.

One important caveat to our study is the finding that mating had no effect on survival or bacterial load in THAI strain females infected with a moderate dose of *S. marcescens*. This experiment was independent from the LVP experiments and utilized a different injection method, thus limiting its informative potential. However, we felt it important to disclose these findings, as they raise the possibility that the effect of mating on immune defense could vary by mosquito strain. An additional consideration highlighted by our experiments is that of infectious dose. As seen in our results, dose influenced the intensity or presence of an effect altogether and should therefore be considered carefully in any experiments regarding pathogenic infection. The insect immune system does not function in isolation as a standalone process within the body. For this reason, immune defense must be considered as something which may be influenced by other physiological processes such as reproduction, and environmental factors such as diet. Here, we have shown that both mating and subsequent blood feeding impact female immune defense, but in different ways. Continued investigation of factors outside of the canonical immune system is critical to develop a thorough understanding of *Ae. aegypti* immune defense.

## Supporting information

**S1 Protocol. Methods for THAI strain experiments explaining any deviations from the protocol used for LVP strain experiments.**
(DOCX)

**S1 Fig. Survival, bacterial load, and presence absence figures for THAI strain experiments.**
(DOCX)

**S2 Fig. Figures to verify the method of virgin collection in this manuscript.**
(DOCX)

**S3 Fig. Figures to verify the method of mating in this manuscript.**
(DOCX)

**S1 Dataset. Complete raw data used for all statistical analyses.**
(XLSX)

**S1 Code. All R code used to produce statistical analyses.**
(R)

**S2 Code. All R code outputs for each experiment providing p-values.**
(DOCX)

**S2 Dataset. Dosing data for each experiment.**
(XLSX)

## Acknowledgments

We thank Noha K. El-dougdoug, Luis E. Martínez Villegas, and James Radl for assistance with bioassays and helpful feedback throughout the project. We also thank Laura Harrington at Cornell University for sharing THAI strain *Ae. aegypti*.

## Author contributions

**Conceptualization:** Brendan J. Kelly, Dom Magistrado, Sarah M. Short.

**Data curation:** Brendan J. Kelly, Nicki C. Joseph.

**Formal analysis:** Brendan J. Kelly, Nicki C. Joseph, Sarah M. Short.

**Funding acquisition:** Sarah M. Short.

**Investigation:** Brendan J. Kelly, Nicki C. Joseph.

**Methodology:** Brendan J. Kelly, Nicki C. Joseph, Dom Magistrado, Sarah M. Short.

**Supervision:** Sarah M. Short.

**Writing – original draft:** Brendan J. Kelly, Nicki C. Joseph, Sarah M. Short.

**Writing – review & editing:** Brendan J. Kelly, Nicki C. Joseph, Dom Magistrado, Sarah M. Short.

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
