## [Decision Letter · Decision Letter 0]

17 Apr 2025

PNTD-D-25-00370

The effects of mating and blood feeding on the immune defense of female Aedes aegypti mosquitoes

Dear Dr. Short,

Thank you for submitting your manuscript to PLOS Neglected Tropical Diseases. After careful consideration, we feel that it has merit but does not fully meet PLOS Neglected Tropical Diseases's publication criteria as it currently stands. Therefore, we invite you to submit a revised version of the manuscript that addresses the points raised during the review process.

Please submit your revised manuscript within 60 days Jun 16 2025 11:59PM. If you will need more time than this to complete your revisions, please reply to this message or contact the journal office at plosntds@plos.org. Please include the following items when submitting your revised manuscript:

We look forward to receiving your revised manuscript.

Kind regards,

Adly M.M. Abd-Alla, Prof asso.

Section Editor

Adly Abd-Alla

Section Editor

Shaden Kamhawi

co-Editor-in-Chief

Paul Brindley

co-Editor-in-Chief

**Journal Requirements:**

1) We do not publish any copyright or trademark symbols that usually accompany proprietary names, eg ©,  ®, or TM  (e.g. next to drug or reagent names). Therefore please remove all instances of trademark/copyright symbols throughout the text, including:

- ® on pages: 7, 8, 9, 10, and 12.

3) Please ensure that the funders and grant numbers match between the Financial Disclosure field and the Funding Information tab in your submission form. Note that the funders must be provided in the same order in both places as well.

**Reviewers' Comments:**

Reviewer's Responses to Questions

**Key Review Criteria Required for Acceptance?**

**Methods**

-Are the objectives of the study clearly articulated with a clear testable hypothesis stated?

-Is the study design appropriate to address the stated objectives?

-Is the population clearly described and appropriate for the hypothesis being tested?

-Is the sample size sufficient to ensure adequate power to address the hypothesis being tested?

-Were correct statistical analysis used to support conclusions?

-Are there concerns about ethical or regulatory requirements being met?

Reviewer #1: Even if I consider the topic of interest and scope of the journal, I do not believe the study design was appropriate to address the objectives. As per the stated objectives and article title, the authors intend to study the "immune defense" in the selected strains of mosquitoes. None of the experiments presented actually shows or includes an actual evaluation of the immune effectors in the mosquitoes (AMP expression, hemocyte evaluation, melanization, ROS responses, etc). Measuring survival and infection prevalence are indirect ways to observe the result of an infection, but it does not by itself means you have studied the immune response of the organism.

The population was clearly described, but not necessarily appropriate for the hypothesis being tested. Collecting adult females from cages with males, even if less than 14 hours old, does not guarantee that mating had not occurred. When working with Aedes mosquitoes, it is common practice to separate pupae to allow female mosquitoes emerge without any contact with males, and doing this would have been easier than separating adults. If a statistical sample would have been selected to check the spermatheca contents either by direct observation or by PCR, and absence of mating would have been shown, this would have been also acceptable even if not ideal.

The justifications for the selection of the mosquito strains used in the study is also not obvious. Why was the THAI strain included only in experiment 3 & 4 and not the others? Was there any indication that any of the variables to be studied (i.e. immune responses, mating, longevity, etc.) could potentially be different between strains? A reader with knowledge about the THAI mosquito strain would know that this colony has been supplemented annually with F1 eggs from the field, and therefore bares more resemblance to field strains than the LVP strain. Yet, this is not mentioned anywhere in the manuscript and therefore we don't know if this was the reason to include this strain in the study or not.

Sample size was sufficient for the experiments presented and I find no ethical concerns with the work.

Reviewer #2: 1.The terms “bacterial load” and “infection prevalence” appear repeatedly across multiple sections and are somewhat redundant in meaning. We recommend merging or simplifying these terms for clarity. Additionally, many sentences in the Introduction are overly nested; these should be revised for conciseness.

2.The number of keywords is excessive; the authors should consider reducing and refining them.

3.As Aedes aegypti is a primary vector for yellow fever virus, the authors should clarify why they focused on bacterial infection instead of the effects on viral transmission (e.g., dengue or Zika virus).

4.The Introduction does not explain the rationale behind choosing Serratia marcescens as the bacterial pathogen. The authors should provide justification for this selection. Furthermore, the rationale for selecting the high, medium, and low infection doses requires support from relevant literature.

Reviewer #3: 1. It is not clear why the authors used two different strains of Aedes aegypti. Specifically why was one strain (THAI) only used for one of the experiments.

2. The description of the experimental design is confusing. It would be helpful to have a experimental design figure to help clarify the differences between the different experiments.

**Results**

-Does the analysis presented match the analysis plan?

-Are the results clearly and completely presented?

-Are the figures (Tables, Images) of sufficient quality for clarity?

Reviewer #1: The results are clearly presented. The figures that present survival data should include the CI. I am concerned with the number of data points presented in figure 1B, considering the number of mosquitoes included, and the number of replicates that the authors made, there should be more points in the 7 day virgin NBF group.

Reviewer #2: 1.The authors are encouraged to assess changes in key immune markers within the mosquito system, such as antimicrobial peptide (AMP) expression or Toll/IMD signaling pathways.

2.The claim that blood feeding impairs immune defense is attributed to egg production or hormonal fluctuations, but these explanations lack direct experimental evidence.

3.All treatment groups were injected with S. marcescens, and no uninfected (healthy) control group was included. The dose ranges used (30.9±11.8 to 184.6±36 CFU) are quite wide, and there is considerable variability in actual doses delivered. Only the immune responses within a 12–48 hour window post-injection were assessed, leaving early or late-phase immune responses unexamined.

4.Flow cytometric analysis of hemocyte proliferation should be included, and critical immune indicators such as antimicrobial peptides and phenoloxidase were not evaluated.

5.The term “significant” is frequently used without providing exact p-values or confidence intervals, making the results interpretation vague.

6.The Results section should be divided into clearly structured subsections to improve the logical flow of the manuscript.

7.Authors should be required to submit color versions of the figures for better clarity.

8.Statistical significance indicators (e.g., p-values, asterisks, confidence intervals) are missing from some of the figures and should be added for transparency.

Reviewer #3: 1. For all figures, significant differences should be made clear with asterisks.

2. Fig. 1C and Fig. 3C & F The exact sample sizes with numerator and denominator should be included in the figure or legend.

3. The figure legends in general rehash what was written in the results section. The figures should be labeled well enough to be self-explanatory, but the legend should be there to help explain the figure. The figure legends should be re-written so as to not re-state the findings but rather explain the figure.

4. Ln. 327 The word prevalence is repeated and one should be removed.

5. Ln. 267 and Fig. 1 legend the authors state one of their analyses was marginally significant. It’s either significant or not. Marginally should be removed as a descriptor.

**Conclusions**

-Are the conclusions supported by the data presented?

-Are the limitations of analysis clearly described?

-Do the authors discuss how these data can be helpful to advance our understanding of the topic under study?

-Is public health relevance addressed?

Reviewer #1: Based on the comments made in the methods section, I do not believe the conclusions are supported by the data presented. The absence of any actual evaluation of the immune effectors of the mosquitoes impairs drawing a conclusion about "immune defenses" being affected or not by mating. Mere survival and infection prevalence are not sufficient to make this statement. The title also ends up being misleading, as it not only indicates that mating affects the immune defense, but that blood feeding also does it, when most of the authors own results end up showing that blood feeding in "vir" vs mated individuals have not difference.

Reviewer #2: (No Response)

Reviewer #3: 1. The authors find that mating leads to increased survivorship yet higher bacterial counts with very significant differences. These seems counter intuitive yet they did not discuss this discrepancy at all. Further, the authors suggest that mating could bolster the immune response yet if this were the case then you would expect reduced bacterial counts. The authors did not provide any evidence that the immune response was altered after mating. To make such claims additional data examining the expression levels of the Toll/ IMD pathways and/ or antimicrobial peptides would be needed. This also applies to the title. They suggest immune defenses in the title but no data directly supporting such claims.

2. Ln. 397 Starting a paragraph with “For example” is kind of odd. I would suggest re-writing this transition.

**Editorial and Data Presentation Modifications?**

Reviewer #1: Page 3, line 58 – It is stated that mating affects lifespan, does it increases it? Or reduces it? Just by the sentence this is not clear.

Page 3, line 59 – but non IN no NBF individuals.

Page 3, line 60-61 – The reference “Change et al., 2021” is not included in the bibliography. ChatGpT? And there are several possible articles to be cited on this topic that have not been included as reference for this statement.

Page 3, line 62—An article whose primary focus is the effects of mating on the immune system could expand a little bit on the literature about the topic in the introduction, especially because in this sentence it is stated that the subject is “many other insects”, citing a single review (Oku et al. 2019) for this seems a little bit lazy.

Page 3, line 64 – If you consider previous work published in Aedes aegypti in mating effects over immunity, including but not limited to: Alfonso-Parra, 2016; Camargo, 2020; Meng-Meng Chang, 2021; Reitmayer, 2021; League, 2021; Taracena-Agarwal, 2024; I would say that the mating effect on immune defense in Ae. Aegypti is not “largely unexplored”. Rather, the mechanisms determining the immune modulation caused by the different components of the mating event need to be better understood.

Page 4, line 77 – reference is needed right after “post-mating”.

Page 5, line 96 – if a careful literature review is done, this sentence would not be included in the manuscript, mating impact in systemic immune defense has clearly already been investigated in Ae. Aegypti.

Page 6, line 117 – no need to cite the same reference in two consecutive sentences.

Page 6, line 122 – I believe it is important to add that you studied these conditions in “young” females, at a “short” time of exposure between the events, with varying doses of Serratia.

Page 6, 128 – This section needs work to be complete as a “experimental design” outline. For the 4 types of experiments performed, it would be useful if you would first denominate the function or intent of each experiment before detailing it. Like: High dose experiment, low dose experiment, blood feeding experiments, etc. Then, it is necessary to add the ages the mosquitoes had, times for the mating, when the injections were performed (and the reasoning behind the selection of these times), and how long after injection the blood meals were performed. It is also not clear to me why only experiment 4 includes the THAI strain; this should be explained as part of the experimental design. Why the larval densities were different in experiment 4 should also be explained.

Page 7, line 134 – Changes in mosquito larval densities can induce significant changes in adults' nutrient reserves. As it happens in lower density trays, mosquitoes reared with more food availability can live longer, endure more environmental pressures, or potentially be more fit for mating. If there were no changes in these parameters with the changes in rearing conditions, this information should be added as supplemental information. If the tray size, water volume, and food mass were adjusted to be kept stable through the change in the number of larvae per tray, this should also be written in the methods.

Page 7, line 150 – what Roth describes in the 1948 paper is that males are not interested in “very young females”, 2-3 hours old, and even with these females they did attempt to mate on a small percentage of success. This reference also describes that it takes less than a minute to mate and other great many details of the mating event. Additionally, it is well established that males emerge earlier than female pupae, by times ranging up to 2 days, so it is mostly always that females will encounter mature males upon emergence. With all of this in mind, the protocol in which the “virgin” mosquitoes were selected is not really one that can guarantee the absence of mating. Collecting virgin adults from a cage, after 12 or 14 hours will very probably have a very significant percentage of mated females already. Therefore, and without any confirmation of mating status by the authors (either by direct observation of the spermathecae or by PCR), I can not say I trust their selection of “virgin” individuals and makes it hard to trust the data corresponding to this group in the manuscript.

Page 7, line 156 – regarding the mating procedure, this is also not an ideal protocol for a study of “mating effects”, which have a known temporal evolution. If the mating is just “assumed” by the mere presence of males being placed in the cage, some females could be mated immediately after the exposure and other could be mated over a day and a half or more later (depending on their experiment number), and at the dissection time there is no way of knowing which female just copulated and which did it a day ago. In this particular experimental design, I even wonder if the females were already mated and could have been refractory by the time of this selected exposure with the males.

In the case where the mating event is the main variable, investing an hour or so to observe the cages while the males are introduced and carefully aspirating the females you actually see copulating is the best guarantee that you have synchronously selected mated females for your study. This, or at the time of dissection, saves the spermathecae of the individuals and confirms sperm presence.

Page 8, line 171 – It should be defined if the females removed were only the ones that did not eat at all and partially engorged females were included in the study or not.

Page 9, line 191 – Even in the Khalil paper, where the methodology is described, it is stated and discussed that the flaw in the method is that it does not allow precise control of the inoculated bacterial load. I am having trouble understanding why experiments 1 to 3 were pricked and the experiment with THAI was injected. Seems to me, that having access to a Nanoject II, there is no obvious reason as of why some experiments would be done with one method over the other. It would be nice to have an explanation of this over in the experimental design section.

Page 9, line 207 – I understand there were experiments done where infected females were blood fed after the infection. It should be presented here, what was the timeline for this series. The authors should state at what age mosquitoes were mated, when they were infected, and then at what age they were blood-fed. If injections affected the volumes of blood acquired, this should also be recorded to guarantee that the differences observed were not actually due to variations in the quantities of blood acquired.

Page 12, line 264 – confidence intervals for the data on figure 1A should be added to the figure. These would be especially important for understanding the “marginally significant” interaction found.

Page 12, line 271 –In Figure 1B, it seems to me that the 7-day virgin NBF has only three points. It is not indicated how many replicates were done for this data set in particular. If it were the same number as for panel 1A, was there one experiment where there were no survivors at this time? If this is the case, this would not be inferred from Figure 1A. More importantly, I believe that the bacterial load data of the blood-fed females could have been an extremely important data to have. Considering the exponential growth of bacteria in the mosquito gut during the blood digestion and the immune modulations known to occur during this time, attributing the deaths to the experimental infection or not can’t be done cleanly without knowing the bacteria load in these individuals. Maybe the Serratia growth didn’t burst PBM but facilitated the leakage of other bacteria from other bacteria.

Page 13, line 284 – with the degree of variation described in the pricking method used for this experiment, the difference between the two infecting doses does not look significant enough to be able to identify dose-dependent effects.

Page 17, line 343 – The format of the legend for the different panels is not consistent with the one used for the other figures.

Reviewer #2: (No Response)

Reviewer #3: (No Response)

**Summary and General Comments**

Reviewer #1: I would like to commend the authors for the work done and I do believe it is a very relevant field of study, with many questions still needing further exploration. I have explained my reservations about the methodology and I hope they are seen as they were intended: constructive criticism. The writing in the paper is in general very clean and well redacted, but I would suggest a deeper search into the literature up to date, to really put together what are the gaps of knowledge regarding the links between mating and immunity in Aedes aegypti.

To claim that mating affects "immune responses", immune responses must be studied, and not only survival or infection prevalence. While I see the benefits of working with whole organisms, it is important to understand that signals triggered by mating affect different tissues in different ways and at different times. Within a same tissue, different cell types could potentially be differentially modulated by different signals coming from the mating event. To unequivocally say that mating impacts immune responses, and in a different way as previously described in many previous of studies, experiments including different components of the immune responses must be made, and the experimental groups must be prepared to guarantee beyond the shadow of the doubt that the virgin females are really it.

Reviewer #2: This study investigates the effects of mating and blood feeding on the immune defense of female Aedes aegypti mosquitoes. The authors found that mating increases survival and bacterial load under medium to high doses of bacterial infection, whereas blood feeding consistently reduces survival across all infection doses.

Reviewer #3: Previous studies in other insect systems have shown that mating can impact the hosts response to infection. In some systems this effect is beneficial while in others detrimental. This has not been empirically tested in Aedes aegypti which is significant because blood feeding is intimately linked to reproduction and blood feeding has been shown to alter mosquito responses. The authors tested whether mating and/ or blood feeding had a significant effect on infection outcomes after exposure to different doses of a pathogenic bacterium, Serratia marcescens. Overall, the experimental design was a little convoluted and could have been better explained, possibly through the addition of a experimental design figure. The analysis of the results seems appropriate but the presentation of the data in the figures could be improved, specifically adding significance demarcations and sample sizes. The finding that mating increased survivorship was significant and adds to the literature; however, the title and discussion are focused on immunity yet there is no data supporting the claims that immunity is important in this phenotype. Further, the authors present conflicting data, i.e. increased survivorship in mated individuals but also significantly higher bacterial counts, which argues against an immune-based mechanism. These contradictions are not discussed. Lastly, in order to draw links to the immune response additional data is needed, the very least being gene expression profiles of antimicrobial genes.

PLOS authors have the option to publish the peer review history of their article (what does this mean? ). If published, this will include your full peer review and any attached files.

**Do you want your identity to be public for this peer review?** For information about this choice, including consent withdrawal, please see our Privacy Policy .

Reviewer #1: No

Reviewer #2: No

Reviewer #3: No

**Figure resubmission:**
---

## [Decision Letter · Decision Letter 1]

9 Sep 2025

Dear Dr Short,

We are pleased to inform you that your manuscript 'The effects of mating and blood feeding on the immune defense of female Aedes aegypti mosquitoes' has been provisionally accepted for publication in PLOS Neglected Tropical Diseases.

Best regards,

Adly M.M. Abd-Alla, Prof asso.

Section Editor

Adly Abd-Alla

Section Editor

Shaden Kamhawi

co-Editor-in-Chief

Paul Brindley

co-Editor-in-Chief

Reviewer #2:

Reviewer #3:

Reviewer's Responses to Questions

**Key Review Criteria Required for Acceptance?**

**Methods**

-Are the objectives of the study clearly articulated with a clear testable hypothesis stated?

-Is the study design appropriate to address the stated objectives?

-Is the population clearly described and appropriate for the hypothesis being tested?

-Is the sample size sufficient to ensure adequate power to address the hypothesis being tested?

-Were correct statistical analysis used to support conclusions?

-Are there concerns about ethical or regulatory requirements being met?

Reviewer #2: (No Response)

Reviewer #3: The authors improved the materials and methods section to make it easier to follow and provided a new figure outlining the experimental approach.

**Results**

-Does the analysis presented match the analysis plan?

-Are the results clearly and completely presented?

-Are the figures (Tables, Images) of sufficient quality for clarity?

Reviewer #2: (No Response)

Reviewer #3: In the new format, the results are clearly and completely presented.

**Conclusions**

-Are the conclusions supported by the data presented?

-Are the limitations of analysis clearly described?

-Do the authors discuss how these data can be helpful to advance our understanding of the topic under study?

-Is public health relevance addressed?

Reviewer #2: (No Response)

Reviewer #3: (No Response)

**Editorial and Data Presentation Modifications?**

Reviewer #2: (No Response)

Reviewer #3: (No Response)

**Summary and General Comments**

Reviewer #2: (No Response)

Reviewer #3: The authors clearly spent considerable time and effort addressing the reviewers concerns. While I don't personally like the use of "immune defenses" in the title and throughout, the science is sound and others working in this space often use this language. I don't have any additional concerns.

PLOS authors have the option to publish the peer review history of their article (what does this mean? ). If published, this will include your full peer review and any attached files.

**Do you want your identity to be public for this peer review?** For information about this choice, including consent withdrawal, please see our Privacy Policy .

Reviewer #2: No

Reviewer #3: No

---

## [Editor Report · Acceptance letter]

Dear Dr Short,

We are delighted to inform you that your manuscript, "The effects of mating and blood feeding on the immune defense of female *Aedes aegypti* mosquitoes," has been formally accepted for publication in PLOS Neglected Tropical Diseases.

Best regards,

Shaden Kamhawi

co-Editor-in-Chief

Paul Brindley

co-Editor-in-Chief
